# Effect of Long-Term Thermal Relaxation of Epoxy Binder on Thermoelasticity of Fiberglass Plastics: Multiscale Modeling and Experiments

**DOI:** 10.3390/polym14091712

**Published:** 2022-04-22

**Authors:** Maxim Mishnev, Alexander Korolev, Bartashevich Ekaterina, Ulrikh Dmitrii

**Affiliations:** 1Department of Building Construction and Structures, South Ural State University, 454080 Chelyabinsk, Russia; 2Research Laboratory of Multiscale Modelling of Multicomponent Functional Materials, South Ural State University, 454080 Chelyabinsk, Russia; 3Department of Town Planning, Engineering Systems and Networks, South Ural State University, 454080 Chelyabinsk, Russia

**Keywords:** polymer composites, glass fiber-reinforced plastics, modulus of elasticity, thermo-relaxation, glass transition temperature, thermal elasticity, epoxy resin, representative volume element

## Abstract

The work is devoted to the prediction and experimental research of the elastic bending modulus of glass-reinforced plastics with an epoxy matrix on anhydride hardener reinforced with different glass fabrics. Experimental studies have been carried out to assess the effect of thermal relaxation of the polymer matrix structure due to long-term exposure to elevated temperatures (above the glass transition temperature of the polymer matrix) on the GRP elastic bending modulus at temperatures ranging from 25 to 180 °C. It has been shown that due to the thermal relaxation of the polymer matrix structure, the GRP modulus increases significantly at temperatures above 110 °C and decreases slightly at lower temperatures. Using a multiscale simulation based on a combination of the finite-element homogenization method in the Material Designer module of the ANSYS software package and three-point bending simulation in the ANSYS APDL module, the elastic modulus of FRP was predicted concerning the temperature, its averaged structural properties, and thermal relaxation of the polymer matrix structure. We have also carried out the prediction of the temperature dependences of the modulus of elasticity of glass-reinforced plastics on different types of glass fabrics in the range from 25 to 200 °C by using the entropic approach and the layered model.

## 1. Introduction

The volume of the application of polymer composite materials in construction has a big potential for growth due to their unique properties. The perspective sphere of polymeric composite materials’ application is in the construction of gas exhaust ducts for industrial enterprises, which include chimneys, gas ducts, gas purification, and desulfurization systems [1,2,3]. Such structures are subjected to joint impact of mechanical loads, increased temperatures, and aggressive media during operation. Under such operating conditions, structures made of conventional materials (reinforced concrete, carbon steel, bricks) collapse at an accelerated rate [4], which can lead to accidents. One of the most recent vivid examples was the collapse of a reinforced concrete chimney 150 m high in March 2022 in Kazakhstan in the city of Petropavlovsk. That is why the construction of gas exhaust ducts made of polymer composites which are resistant to aggressive influences are a good alternative to the traditional similar constructions.

Glass plastics based on epoxy resins, which have good mechanical characteristics, manufacturability, chemical resistance, and affordable cost, are often used for manufacturing the considered structures. Examples of fiberglass shells of gas exhaust ducts based on epoxy fiberglass plastics are given, for example, in [4]. The available successful examples of the implementation of gas exhaust ducts made of epoxy glass-reinforced plastics are limited to a long-term operation at temperatures not exceeding 120 to 130 °C and many industrial enterprises are interested in the long-term operation of gas exhaust ducts made of polymeric composite materials at temperatures at least 180 to 200 °C. It should be noted that the cost of composites should not increase essentially for the construction to remain economically effective. That is why it is reasonable to estimate the possibility of their production from widespread and relatively inexpensive types of polymeric binders, such as hot-curing epoxy resins with anhydride hardener.

To develop a new generation of gas exhaust ducts’ construction, it is necessary first to provide the creation of composite or hybrid materials that can be operated for a long time at temperatures up to 180 to 200 °C while maintaining good mechanical properties (strength and stiffness) and remain economically efficient to the construction industry.

The thermomechanical properties of fiberglass plastic are determined by the thermomechanical properties of the binder.

Thermal aging of an anhydride-cured epoxy resin in the temperature range 130 to 160 °C was explored in [5]. The effects on the flexural properties, molecular structures, free volume fraction, and mechanical properties were investigated, which noted that it leads to oxidation and molecular chain rearrangement in the skin of the epoxy samples. The flexural test results indicated that thermal aging reduced the breaking strain, while the flexural strength was only slightly affected, and the value of the modulus increased.

The thermal–mechanical properties and chemical structure of ambient cured DGEBA/TEPA during thermo–oxidative aging at temperatures up to 110 °C were investigated in [6]. It was noted that the glass transition temperature increased with the aging temperature and time because of post-cure reactions under the thermo–oxidative aging process.

Prediction and experimental study of physical and mechanical properties of thermosetting polymer binders at elevated temperatures (including epoxy) were undertaken in previously published articles [7,8]. It was shown that thermal relaxation of the considered polymers structure with mass loss occurred under prolonged exposure to temperatures exceeding the glass transition temperature due to yielding of the most volatile components. For the epoxy binders considered, the loss of mass at temperatures up to 200 °C had a damping character, which indicates the potential for long-term operation at this temperature. It was also shown that after aging, the glass transition temperature of the epoxy binder on the anhydride hardener increased by about 1.3 times, and the modulus of elasticity at elevated temperatures increased by more than 1.8 times. At the same time, the modulus of elasticity at 22 to 25 °C after curing practically did not change.

This suggests that prolonged exposure to elevated temperatures, accompanied by a gradual decaying loss of mass, is associated with processes that reduce the segmental mobility of the polymer at elevated temperatures, which causes the transition to a highly elastic state according to [9].

Thus, the mechanical glass transition temperature and the thermoelastic characteristics of polymeric binders can change significantly under the influence of operating factors, e.g., long-term exposure to temperature. The glass transition temperature immediately after complete polymer curing (before prolonged exposure to any operating factors that could cause a change) can be called the initial glass transition temperature.

The effect of temperature and time of exposure on the impact and flexural mechanical responses of carbon/epoxy composites were studied in [10], which noted that for samples aged at 190 °C, the effect of temperature and time of aging caused a progressive decrease in mechanical properties with an increase in delamination and microcracks, but no sign of a consolidation stage was observed.

The aim of [11] was to apprehend the effect of oxidation on the transverse composite properties directly at the ply’s scale, and to examine its effect on the evolution of transverse cracking during aging and mechanical loading. The experimental results highlighted the presence of an oxidized/degraded layer thinner than the external 90° ply and its role in the development of the kinetics of transverse cracking.

A monograph [12] showed that weight loss and glassing temperature changes are not related to discrete changes in the mechanical properties of composites. Moreover, these changes do not directly reflect different chemical and physical mechanisms that control mechanical–property degradation. Indeed, the various composite properties that are affected by thermo–oxidative aging-induced changes in fiber, matrix, interface properties, or a combination of these, will vary in different ways with aging.

This work aims to predict and investigate the effect of long-term exposure at elevated temperatures exceeding the initial glass transition temperature of epoxy glass-reinforced plastic matrix on its elasticity modulus at temperatures ranging from 22 to 25 °C to 200 °C experimentally. This information is required for further calculations of construction structures made of epoxy FRP materials operating under the combined influence of mechanical loads and temperatures up to 200 °C, such as composite gas ducts and chimneys of metallurgical, or power enterprises.

The initial data for the prediction of the temperature dependence of the elastic modulus of fiberglass plastic under bending, as well as its change due to thermal aging of an epoxy matrix, were the previously obtained [7] temperature dependences of the elastic modulus of an unreinforced epoxy matrix (including those under thermal aging) and averaged parameters of fiberglass plastic specimens’ structure, obtained by optical microscopy data. The prediction was performed using a combination of finite element (FE) homogenization and finite element modeling of three-point bending tests on flat fiberglass specimens.

The deformation characteristics of epoxy-bonded fiberglass plastic were predicted in the academic version of the ANSYS finite-element package using the built-in module Material Designer. The elastic characteristics of the woven composite were calculated based on the finite-element (FE) homogenization method [13,14,15]. In this method, a small representative volume of the material (Representative Volume Element (RVE)) is extracted, which has sufficient dimensions to possess macroscopic characteristics of elastic properties. The prediction of the mechanical characteristics of polymer composites using multiscale modeling and finite-element homogenization was undertaken in [13,14,15,16,17,18,19].

The Material Designer module of Ansys Workbench is intended for solving problems of structural micromechanics. Examples of modeling the mechanical characteristics of a unidirectional composite in the Material Designer module are given in [20].

## 2. Materials and Methods

### 2.1. Materials

In this work, we considered fiberglass plastics made based on structural glass fabrics EZ-200 and T-23 and a hot-cured epoxy binder (EP) with anhydride hardener.

The epoxy binder (EP) for the fiberglass plastic was made based on epoxy resin KER 828 (South Korea); isomethyltetrahydrophthalic anhydride (IZOMTHFA) was used as a hardener; 2,4,6-tris-(dimethylaminomethyl)-phenol, produced under the brand name Alkophen, was used as a curing gas pedal. The weight ratio of the ES components was as follows: KER 828–54.5%, IZOMTGFA—42.5%, Alkophen—3%. The components described below were used to make the binders:Epoxy resin KER 828, with the following main characteristics: Epoxy Group Content (EGC) 5308 mmol/kg, Epoxide Equivalent Weight (EEW) 188.5 g/eq, viscosity at 25 °C 12.7 Pa·s, HCl 116 mg/kg, and total chlorine 1011 mg/kg. Manufacturer: KUMHO P&B Chemicals, Gwangju, South Korea.Hardener for epoxy resin methyl tetrahydrophthalic anhydride with the following main characteristics: viscosity at 25 °C 63 Pa·s, anhydride content 42.4%, volatile fraction content 0.55%, and free acid 0.1%. Manufacturer: ASAMBLY Chemicals company Ltd., Nanjing, China.Alkophen (epoxy resin curing accelerator) with the following main characteristics: viscosity at 25 °C 150 Pa·s, molecular formula C_15_H_27_N_3_O, molecular weight 265, and amine value 600 mg KOH/g. Manufacturer: Epital JSC, Moscow, Russian Federation.

The components were mixed in the above proportions at room temperature of about 25 °C. Mixing to a homogeneous consistency was carried out mechanically with an electric drill with a mixing attachment.

Glass cloth EZ-200 was produced according to the standard GOST 19907-83 [21] and had the following characteristics:-Thickness 0.190 + 0.01/−0.02 mm;-Surface density 200 + 16/−10 g/m^2^;-Number of yarns per 1 cm of fabric on the basis 12 +/− 1;-Number of yarns per 1 cm of fabric on the weft 8 +/− 1;-Weave—plain;-Oiling agent—paraffin emulsion.

Glass fabric T-23 was produced in accordance with GOST 19170-2001 [22] and had the following characteristics:-Thickness, 0.27 + 0.01/−0.02 mm;-Surface density, 260 + 25/−25 g/m^2^;-Number of yarns per 1 cm of fabric on the basis 12 +/− 1;-Number of yarns per 1 cm of fabric on the weft 8 +/− 1;-Weave—plain;-Oiling agent—aminosilane.

Samples of fiberglass plastic were made in the form of plates of 15 cm × 15 cm. Cut sheets of glass fabric EZ-200 were calcined at 300 °C to remove the paraffin oiling agent immediately before impregnation with the binder. Glass fabric T-23 was not calcined. In total, the samples had 10 layers of glass fabric laid according to the scheme 0/90 (base/weft).

Glass-reinforced plastic specimens were cured at 120 °C for 20 min in silicone molds while being loaded through Teflon-coated metal plates at a pressure of about 0.22 kPa. The cured specimens were then kept at 150 °C for 12 h. After that, beam samples were cut from the plates in the direction of the main axes of orthotropy, which were considered in this work.

### 2.2. Methods

#### 2.2.1. Long Heat Treatment

After curing, some of the fiberglass samples were exposed to prolonged exposure at elevated temperatures, while the control series was stored under normal conditions. The long-term curing (hereinafter referred to simply as “curing”) of the samples at elevated temperatures was performed according to the following program: 168 h (one week) at 160 °C, 168 h at 190 °C, 168 h at 220 °C. After the heat treatment, the samples were cooled at a rate of about 1 °C per minute to 50 °C, removed from the laboratory oven, weighed, and then weighed and tested for three-point bending at temperatures from 25 to 180 °C.

#### 2.2.2. Investigation of Elasticity Modulus under Heating

Polymer samples were tested for static three-point bending on a Tinius Olsen h100ku test machine (Tinius Olsen Ltd., 6 Perrywood Business Park, Honeycrock Lane, Salfords (Near Redhill), Surrey RH1 5DZ, England) in a specially made, small-sized chamber that provided heating and maintained the temperature up to 300 °C. Three-point bending tests were carried out according to GOST R 56810-2015 [23]. According to the passport data, the load measurement accuracy of the Tinius Olsen h100ku machine is ±0.5% in the range from 0.2% to 100% of the allowable load of the installed force sensor (100 kN). The crosshead has a resolution of 0.001 mm with an accuracy of 0.01 mm. To eliminate the influence of machine stickiness, the displacement of the specimen center point under load was also controlled by a mechanical watch type indicator mounted under the specimen. The difference in displacement readings on the crosshead and the dial indicator did not exceed 2%. The specimens were tested at a span of 70 mm.

To determine the modulus of elasticity and bending strength at room temperature, six samples were tested on different types of glass fabrics aged and not aged at elevated temperatures (a total of 24 samples were tested). Appearance of fiberglass specimens after prolonged exposure at elevated temperatures, specimens prepared for exposure, and three-point bending process at room temperature showed on the Figure 1.

To determine the temperature dependence of bending modulus, two samples were tested on different types of glass fabrics kept at elevated temperatures and not kept at elevated temperatures (a total of 4 samples were tested). For comparison with the results of modeling, one of these four FRP samples was chosen for which geometrical features of the structure (thickness of cross-section and a separate reinforced layer, the distance between the centers of threads, a ratio of the cross-section area of separate fibers to the area of yarn they form, etc.) were determined by optical microscopy and set as the initial data for modeling in the Material Designer module of the ANSYS package.

#### 2.2.3. Prediction of the Deformation Characteristics of FRP

The Material Designer module of ANSYS package automatically builds the bulk FE model of the unit cell and calculates the orthotropic or fully anisotropic elastic characteristics of the homogenized material based on the specified material characteristics of the matrix and filler as a fabric. This method of modeling considers the curvature of the yarns in the fabric structure as well as the influence of the transverse yarns on the elastic modulus in the longitudinal direction.

According to [24], the theoretical basis for the finite-element homogenization used in the Material Designer module is in the works [14,25]. They described the basic principles of RVE and unit cell forming. When modeling woven composites, the following assumptions are used: matrix and fiber materials are linearly elastic; the yarn fiber volume fraction is constant; the weaving pattern is regular, and layers of the woven composite are laying exactly on top of each other.

In the first stage, based on the model parameters entered (see below), an approximate geometric RVE model is created in the Material Designer module, based on which a volume finite-element mesh is constructed. Then, several types of macroscopic loads are simulated to determine the orthotropic elastic characteristics: 3 tensile tests (X, Y, Z) and 3 shear tests (XY, YZ, XZ). A corresponding macroscopic strain is applied in each case, and reaction forces in the boundary faces of the RVE are used to assemble the stiffness matrix. Engineering constants are then extracted.

For example, when simulating a tensile test along the *x*-axis, the boundary conditions are the relative strain in the *x*-axis direction has a fixed value of 0.001; all other relative strains are 0. This allows the first column of the stiffness matrix to be reduced to the following form:$$\left(\begin{array}{c} D_{11}\\ D_{21}\\ D_{31}\\ 0\\ 0\\ 0 \end{array}\right)=\frac{1}{0.001}\times \left(\begin{array}{c} \sigma_{x}\\ \sigma_{y}\\ \sigma_{z}\\ \sigma_{xy}\\ \sigma_{yz}\\ \sigma_{xz} \end{array}\right).$$

Assume the RVE occupies the volume $\left[0,L_{x}\right]\times \left[0,L_{y}\right]\times \left[0,L_{z}\right]$. On the faces normal to the *x*-axis, enforce
$$u_{x}\left(L_{x},y,z\right)=u_{x}\left(0,y,z\right)+\epsilon L_{x}\;u_{y}\left(L_{x},y,z\right)=u_{y}\left(0,y,z\right)\;u_{z}\left(L_{x},y,z\right)=u_{z}\left(0,y,z\right).$$

On the faces normal to the *y*-axis, enforce
$$u_{x}\left(x,L_{y},z\right)=u_{x}\left(x,0,z\right)\;u_{y}\left(x,L_{y},z\right)=u_{y}\left(x,0,z\right)\;u_{z}\left(x,L_{y},z\right)=u_{z}\left(x,0,z\right).$$

On faces normal to the *z*-axis, enforce
$$u_{x}\left(x,y,L_{z}\right)=u_{x}\left(x,y,0\right)\;u_{y}\left(x,y,L_{z}\right)=u_{y}\left(x,y,0\right)\;u_{z}\left(x,y,L_{z}\right)=u_{z}\left(x,y,0\right).$$

In addition to these periodicity conditions, rigid body motions must also be prevented. This is done by enforcing
$$u_{x}\left(a\;point\;with\;x=0\right)=0\;u_{y}\left(a\;point\;with\;y=0\right)=0\;u_{z}\left(a\;point\;with\;z=0\right)=0.$$

To compute macroscopic stresses, the forces on the top faces are integrated. By repeating the steps for all the other load cases, all the entries for the stiffness matrix are obtained. The stiffness matrix is inverted to obtain the compliance matrix: [C] = [D]^−1^.

Finally, the engineering constants are computed from the relationship
$$\left[C\right]=\left(\begin{array}{cccccc} \frac{1}{E_{x}} & \frac{\nu_{yx}}{E_{y}} & \frac{-\nu_{zx}}{E_{z}} & 0 & 0 & 0\\ \frac{-\nu_{xy}}{E_{x}} & \frac{1}{E_{y}} & \frac{-\nu_{zy}}{E_{z}} & 0 & 0 & 0\\ \frac{-\nu_{xz}}{E_{x}} & \frac{-\nu_{yz}}{E_{y}} & \frac{1}{E_{z}} & 0 & 0 & 0\\ 0 & 0 & 0 & \frac{1}{G_{xy}} & 0 & 0\\ 0 & 0 & 0 & 0 & \frac{1}{G_{yz}} & 0\\ 0 & 0 & 0 & 0 & 0 & \frac{1}{G_{xz}} \end{array}\right)$$

The elastic characteristics of glass fiber were used as input data for creating the model in the Material Designer module; the elastic characteristics of the cured epoxy matrix before and after curing were obtained in [7], on which the fiberglass plastic was manufactured, were determined by the three-point bending method. Initial data for constructing geometry, geometric and finite element RVE models showed on the Figure 2.

The characteristics are calculated under the following assumptions:-The fiber composite being modeled consists of isotropic linear-elastic matrix material and isotropic or transversally isotropic linear-elastic filament material;-The volume fraction of fibers in the filaments is constant;-The representative volume of the material is strictly periodic.-The following parameters are set as the initial data for the woven composite model:-Weaving type—a type of fabric weaving (plain or twill);-Fiber volume fraction—share of fiber volume in the volume of the whole RVE;-Yarn fiber volume fraction is the fraction of volume in a separate thread, which is taken up by the fiber material (glass in our case), the volume of the yarn “net”;-Shear angle—the angle in degrees of warping the fibers due to drape properties of the fabric;-Yarn spacing—the distance between the centers of cross-sections of neighboring yarns;-Fabric thickness—thickness of the modeled RVE;-Repeat count—number of elementary cells considered in the model in the direction of each coordinate;-Align with x-direction—the fibers are oriented along the *x*-axis (if not, they are oriented at a 45° angle to the *x*-axis).

As follows from the help system [24], the Fiber volume fraction parameter is defined as the product of the Yarn volume fraction parameter by the Yarn fiber volume fraction parameter (see the transcription above). The Yarn volume fraction is not set directly, it corresponds to the fraction of the “gross” volume of all filaments (without considering the space between individual fibers composing a filament) to RVE volume. The Yarn volume fraction parameter can be defined as the ratio of the “gross” cross-sectional area of the yarns oriented along with the sample to the cross-sectional area of the sample.

The three-point bending of fiberglass specimens, the results of which were compared with the experiment, was simulated in the academic version of the ANSYS APDL package. The specimens were modeled with Shell181 shell elements [26], supporting multilayer sections.

The study of mesostructure features of fiberglass plastics was carried out using Levenhuk DTX 90 and Levenhuk 320 BASE optical microscopes equipped with digital cameras. The Levenhuk DTX 90 microscope was used to clarify the actual thicknesses of the layers in the fiberglass structure and to determine the value of the Yarn volume fraction parameter (the proportion of the “gross” cross-sectional areas of the yarns from the cross-sectional area of the element). Levenhuk 320 BASE microscope with 40:1 magnification in transmitted light was used to determine the Yarn fiber volume fraction parameter.

The obtained digital photos of the structure of fiberglass samples were imported into the NanoCAD SPDS package (version for educational institutions) and brought to the same scale. The thickness of the sample at the marked point was taken as the base size, which was measured with a caliper with an accuracy of 0.05 mm. After reduction to the same scale, the thickness of layers, shape, and size of yarns, shares of space occupied by yarns inside samples (Yarn volume fraction), shares of space occupied by fibers inside yarns (Yarn fiber volume fraction) were estimated. The RVE geometric model image from the Design Modeler module was imported and scaled for comparison with the real structure of the samples.

In this work, the task was to predict by numerical modeling the characteristics of a particular sample, considering its individual structural features, so statistical processing was not required and just one sample was considered.

The density of samples before and after exposure at elevated temperatures was determined by hydrostatic weighing.

## 3. Results

### 3.1. Experimental Results

Three-point bending tests to determine the modulus of elasticity were conducted at room and elevated temperatures. In the first stage, tests were performed and the dependence of modulus of elasticity on temperature was plotted for samples not subjected to long-term exposure at elevated temperatures after the same samples were exposed to temperatures of 160, 190, and 220 °C (see Section 2.2) and a new dependence of the modulus of elasticity of fiberglass plastic on temperature was plotted, taking into account changes in the rigidity of epoxy matrix as a result of long-term exposure to elevated temperatures. Three-point bend tests were performed at room temperature on fiberglass samples cut from the same plate to determine durability; some samples had been exposed to temperatures of 160, 190, and 220 °C previously. Fiberglass specimens based on EP epoxy binder and EZ-200 and T-23 glass fabrics were tested (see Section 2.1).

The results of experimental determination of temperature dependences of the modulus of elasticity of glass-reinforced plastics on different types of glass fabrics before and after exposure at elevated temperatures are shown in Figure 3 and Figure 4.

The results of experimental studies showed that the modulus of elasticity of glass-reinforced plastics after long-term exposure at elevated temperatures (above the initial glass transition temperature) increased significantly at temperatures above 110 °C and decreased slightly at temperatures below 110 °C. For example, at 140 °C, the elastic modulus after curing increased by a factor of 1.4 to 1.6 for samples based on the EZ-200 glass fabric, and by a factor of 1.7 to 1.9 for samples based on the T-23 glass fabric.

The increase in the modulus of elasticity of FRP at elevated temperatures is due to an increase in the glass transition temperature and the modulus of elasticity of the epoxy polymer because of the thermal relaxation of its structure after exposure at temperatures higher than the initial glass transition temperature, as was shown in [7].

The results of determining the bending strength and modulus of elasticity at room temperature of fiberglass plastics before and after curing are presented in Table 1. Columns 11 and 12 show the average values of elastic modulus and bending strength at room temperature; the values of standard deviations are shown under the slash, and the values of coefficients of variation in percent are shown in brackets. Figure 5 shows the average values of bending modulus (a) and bending strength (b) with standard deviations for fiberglass plastics based on EZ-200 and T-23 fabrics before and after aging at temperature.

According to the results obtained, the bending strength after aging decreased on average by 24 and 26% in fiberglass plastics based on EZ-200 and T-23 fabrics, respectively, and the modulus of elasticity decreased by 18 and 6%, respectively.

The table (Table 2) shows the averaged results of density change and weight loss of the tested FRP specimens after exposure.

### 3.2. Prediction of the Thermomechanical Characteristics of Fiberglass Plastics Using the Finite Element Method

The accurate prediction of the modulus of elasticity of FRP in a wide temperature range, including considering its change after a long exposure at elevated temperatures, is necessary to assess the stress–strain state of structures during operation. The structure of glass-reinforced plastics may vary depending on the materials and production technology used, which should be considered in the modeling.

In this paper, the thermomechanical properties of glass-reinforced plastics have been modeled considering changes in the temperature dependence of the elastic modulus of the epoxy matrix after long exposure at temperatures higher than the initial glass transition temperature and considering individual structural features of the sample.

A fiberglass plastic sample number 1-1 based on the epoxy binder and EZ-200 glass fabric was chosen for modeling. Its structure was investigated and recorded using digital microscopes (see Section 2.2), and then the geometric parameters of the structure were set as input data for the finite-element calculation in the academic version of the ANSYS package using the Material Designer and ANSYS APDL modules. A microphotograph of the cross-sectional fragments of the sample is shown in Figure 6 and Figure 7.

As shown in the paper [27], for correct modeling of the deformation of glass-fiber-reinforced plastics, it is recommended to perform it in two stages using a structural–phenomenological model reflecting (although simplified) the actual features of the mesostructure of a specific glass-fiber-reinforced plastic:-Stage 1—determination of the elastic modulus of a reinforced monolayer based on the finite-element (FE) homogenization method [13,14] (for example, in the Material Designer module of the ANSYS Workbench package);-Stage 2—the refinement of the elastic modulus of the multilayer composite using three-point bend modeling (for example, in the ANSYS APDL module), considering the polymer matrix interlayers between the reinforced monolayers.

To estimate the average “gross” area of yarn in the cross-section, a rectangular section of 2.45 mm × 4.0 mm was isolated, and the average “gross” area of yarn was determined as the arithmetic average of the total area of all yarns that were entirely inside this section. The average gross area per yarn determined in this way was 0.0626 mm^2^.

The value of the parameter Yarn volume fraction determined as a ratio of the total “gross” area of all yarns to the area of the selected rectangular area of 2.45 mm × 4.0 mm was 0.253.

Figure 7 shows the actual distances between the centers of the longitudinal yarns; the average value of these distances was 0.98 mm and was taken as the value of parameter Yarn spacing.

Figure 6 shows a transmitted-light microphotograph of a cross-sectional fragment showing longitudinal and transverse filaments; individual glass fibers can be distinguished in the cross-section of the longitudinal filaments. The diameter of the individual glass fibers measured from the scaled microphotograph was about 0.008 mm (8 µm). To determine the fraction of fiber area within the “gross” area of a filament, a shaded circle of 0.008 mm (8 µm) diameter was applied to each fiber, after which the value of the Yarn fiber volume fraction parameter was determined as the ratio of the total area of shaded circles to the “gross” area of the filament. The average value of the Yarn fiber volume fraction parameter determined from the four strands shown in Figure 6 was 0.459.

As a result, the value of the Fiber volume fraction was determined as the product of the Yarn fiber volume fraction by the Yarn volume fraction; it was 0.253 × 0.459 = 0.1161.

In the Material Designer module, the mechanical properties of the epoxy matrix were specified as input data (Figure 8) before and after exposure to elevated temperatures, as defined in [7]; the mechanical characteristics of glass fiber were based on the E-Glass material available in the ANSYS library.

As a result, after performing calculations in Material Designer, the temperature dependences of the elastic modulus of the reinforced fiberglass monolayer were obtained as shown in Figure 9. According to the obtained results, we see that the modulus of elasticity of FRP before and after exposure at elevated temperatures, calculated for the monolayer by the finite-element method, differed significantly from the experimental value, as well as in the character of the temperature dependence. For example, at room temperature, the calculated monolayer elastic modulus exceeded the experimental one by a factor of about 1.4. As was shown in [27], the results using the orthotropic model (i.e., independent shear modulus values), as well as an increase in the number of considered elementary cells, did not differ significantly.

The finite-element (FE) model (Figure 10) for elastic modulus prediction consisted of alternating reinforced monolayers (10 pieces), layers of unreinforced epoxy matrix between them (11 pieces), and one thickened bottom layer of epoxy matrix, which corresponded to the actual sample geometry. Temperature-dependent elastic characteristics of the reinforced monolayers were taken from the results of prediction based on the finite-element (FE) homogenization method in the Material Designer module; elastic characteristics of resin interlayers were taken from the experiment.

The finite-element (FE) model of sample 1-1 was a multilayer plate hinged at the ends (one end was fixed at three spatial coordinates, the other end was fixed only at the vertical coordinate), and a force of 100 N was applied to the center of the sample. From the results of calculating the deflection of the sample at different temperatures, we calculated the value of the reduced modulus of elasticity of the sample to compare it with the experimentally obtained values. Finite element calculations were performed in a linear and geometrically nonlinear formulation.

The thickness of reinforced monolayers was taken as 0.185 mm. The thickness of the lower resin layer (which appeared thicker by microscopy) was 0.201 mm on average; the thickness of resin layers between reinforced monolayers was taken as equal and equal to (2.45 − (0.185 × 10 + 0.201)/11 = 0.036 mm.

The results of the determination of the calculated modulus of elasticity of the considered glass-reinforced plastic sample in the temperature range of 25 to 180 °C are shown in Figure 11. According to the obtained results, prediction of the modulus of elasticity of fiberglass plastic for practical tasks with the use of the structural–phenomenological model gives much better qualitative and quantitative coincidence with the experimental results than prediction using only the finite-element homogenization method using RVE in the module Material Designer. Thus, at a temperature of 25 °C, an almost complete coincidence with the experiment was obtained. The results of the geometrically nonlinear calculation at elevated temperatures showed values closer to the experiment than the results of the linear calculation by about 20%.

At elevated temperatures, the experimental values of the modulus of elasticity of fiberglass samples exceeded the predicted calculated values, which on the one hand, requires additional research to clarify the reason for the difference; on the other hand, in terms of reliability for solving practical problems, it is better than overestimating the modulus of elasticity, as it goes into the reserve of reliability.

The percentage prediction error for different temperature values of the modulus of elasticity of FRP relative to the experimental values is shown in Figure 12.

### 3.3. Predicting Properties Using the Entropy Approach

Let us carry out a comparative analysis of the dependences of the elastic modulus of polymers and fiberglass plastic based on them. Experimental dependences of the modulus of elasticity of fiberglass plastics on temperature are shown in Figure 3, experimental dependences of the modulus of elasticity of unfilled epoxy polymer on temperature are shown in [7,27].

Consideration of the dependencies made it possible to conclude that the modulus of elasticity changes upon heating in polymers and glass-reinforced plastics is very similar, both before and after prolonged exposure at elevated temperatures. Heat treatment significantly increases the elastic properties at elevated temperatures and the modulus of elasticity of the glass-reinforced plastics is several times higher than that of cured unreinforced polymers.

As shown above, for predicting the thermoelastic properties of glass-reinforced plastics, the structural-phenomenological multilayer model of the plastic, in which two elements are distinguished—the reinforced monolayer directly containing the fiber, and the unreinforced polymer layers linking the reinforced layers.

Therefore, when modeling the temperature dependence of the GRP elasticity, it is proposed to consider the GRP as a hybrid multilayer material (macro polymer) in which the number of elastic bonds in the matrix layer between the elastic reinforced layers decreases when heating by analogy with modeling the deformation properties of the polymer when heating. The proposed model is shown in Figure 13.

Considering similar dependencies of hardened polymers and fabric-reinforced composites, as well as the multilayer structure, we used the previously obtained and proven dependence of the elastic modulus of hardened polymer on temperature:(1)$$E_{T}=\frac{k_{pl}E_{0}}{1-n_{T}(1-k_{pl})}.$$

$n_{T}$—share of elastic bonds in the total number of bonds;

$k_{pl}$—is the ratio of the strain modulus of plastic bonds to the elastic modulus of elastic bonds (coefficient of deformability).

Based on previous studies of polymer structure with temperature change [7,8], the coefficient of deformability of adsorption bonding is related to temperature by the entropy equation
(2)$$k_{pl}=1-S\times ln\frac{T_{s}}{T_{0}}.$$

S—[J/J]—coefficient of bond entropy, equal to the ratio between the entropy and the potential energy of elastic bonding at a standard temperature.

S—correlation parameters with the physical essence at this stage of the study.

$n_{T}$ = 0.71 we take as the share of elastic reinforced layers in the investigated fiberglass plastic.

Thus, the equation takes the form
(3)$$E_{T}=\frac{k_{pl}E_{0}}{1-0.71(1-k_{pl})}.$$

The obtained entropy coefficients and the results of calculations and experiments are shown in the tables (Table 3 and Table 4).

It follows from the comparative data that while the discrepancy between the calculated model and the experiment for FRP before heat treatment can be quite significant, the data after heat treatment diverged from the experiment by no more than 15%, and in more cases, by no more than 10%.

In general, the entropy value of fiberglass plastic was up to 2 times lower than that of unreinforced polymer, indicating that the composite structure was much more ordered. After thermal relaxation, the entropy level decreased by another 20% and leveled out for both types of FRP (on different types of glass fabrics). Therefore, the difference between the elastic properties of fiberglass composites on different fabrics may be related to their somewhat different fiber–matrix bonding properties. After heat treatment, the bonding strength increased and leveled out, as do the elastic properties of the plastics in general.

It should be noted that thermal relaxation made it possible to maintain significant elastic properties of glass-reinforced plastics on the considered epoxy matrix up to 180 °C. It should also be noted that the most important factor of the elastic modulus at elevated temperature was the content of the polymer matrix and non-reinforced layers. The layered model shown above made it possible to estimate a possible increase in elastic modulus with a decrease in polymer consumption at different temperatures (Figure 14).

The relationship showed that the amount of unreinforced polymer content had a particularly significant effect on the elastic properties of FRP at elevated temperatures. Doubling the amount of unreinforced polymer can also increase the modulus of elasticity of the plastic by about two times and make it usable even at 200 °C, i.e., at temperatures above the glass transition temperature of the polymer by about 60 °C.

## 4. Discussion

Experimental investigations showed that the modulus of elasticity of the glass-reinforced plastics with epoxy matrix examined increased significantly at temperatures above 110 °C and slightly decreased at temperatures below 110 °C after a long period of exposure at elevated temperatures (above the initial glass matrix melting point). For example, at a temperature of 140 °C, the modulus of elasticity after curing increased by a factor of 1.4 to 1.6 for samples based on the EZ-200 glass fabric, and by a factor of 1.7 to 1.9 for samples based on the T-23 glass fabric. The effect of long-term exposure on the bending strength of FRP specimens at room temperature was also evaluated; according to the results obtained, it decreased 1.24 to 1.26-fold.

Thus, it can be stated that long-term exposure at an elevated temperature has different effects on the mechanical properties of the considered epoxy matrix and FRP on a basis depending on the temperature conditions of exploitation, which are often variable in the structures of gas exhaust ducts of industrial enterprises. Changes in thermomechanical properties of fiberglass plastic will essentially influence the formation of the stress–strain state of constructions, their reliability, and durability. That is why the question of reliable prediction of composite properties at long-term influence of high temperatures remains essential.

The prediction of the temperature dependence of the elastic modulus on the example of a single fiberglass plastic specimen, considering its specific structural features and the effect of thermal relaxation of the polymer matrix structure, was performed by using a method based on multiscale modeling with finite-element homogenization and subsequent transition to the structural–phenomenological model of three-point bending. Comparison with the experimental data showed that the calculated results obtained by this method coincided well with the experiment; therefore, the proposed method can be used in practice for predicting the mechanical characteristics of layered composites.

The exceeding of the experimental values of the strain modulus at elevated temperatures (with almost complete coincidence at room temperature) as compared to the calculated values can be due to the following hypothetical reasons.

The study of thermal relaxation of binders in work [7], on the result of which the prediction in the present work was based, was carried out on large-sized samples (thickness more than 10 mm). Perhaps that is why the thermal relaxation of the structure, which determines the increase of rigidity of polymer at increased temperatures, in that large-sized sample, could not be completely realized in their inner volume. This is also indirectly evidenced, for example, by the results of [5], where it was noted that the greatest changes as a result of thermal aging occur in the thin outer layers of the samples. At the same time, in the fiberglass plastic specimens considered in this work, resin layers were much thinner, and thermal relaxation could be completely realized in the whole volume. Therefore, it is advisable to conduct further studies of polymer thermal relaxation considering the scale factor.

Another hypothetical reason may be that in a real composite, the transition of a polymeric binder into a forced elastic state (i.e., a highly elastic state manifested by the combined action of temperature and mechanical stresses) [9] occurs later than in a pure (unreinforced) binder. This question also requires clarification during further studies.

## 5. Conclusions

Among the most interesting results we can distinguish the following:After a prolonged exposure of the fiberglass samples (504 h in total) at temperatures higher than the initial glass transition temperature of the polymer–binder matrix, the flexural modulus at temperatures above 100 to 110 °C significantly increased (up to 1.6 times for samples on the EZ-200 fabric, up to 1.9 times for samples on the T-23 fabric), at lower temperatures the flexural modulus either did not change or decreased slightly.The bending strength after curing decreased by about 16% in the samples on the EZ-200 fabric and by 8% in the samples on the T-23 fabric.The entropy value of fiberglass plastic was up to two times lower than that of unreinforced polymer, indicating that the composite structure was much more ordered. After thermal relaxation, the entropy level decreased by another 20% and leveled out for both types of FRP (on different types of glass fabrics). Therefore, the difference between the elastic properties of fiberglass composites on different fabrics may be related to their somewhat different fiber–matrix bonding properties.The calculated values of the modulus of elasticity at room temperature showed the best coincidence with the experiment (almost complete coincidence), while at elevated temperatures, the experimental values of the modulus of elasticity were higher than the predicted ones, which, on the one hand, requires additional research to clarify the reason for the difference, and on the other hand, from the standpoint of reliability for solving practical problems, it is better than overestimation, since it goes into the reliability reserve.

## Figures and Tables

**Figure 1 polymers-14-01712-f001:**
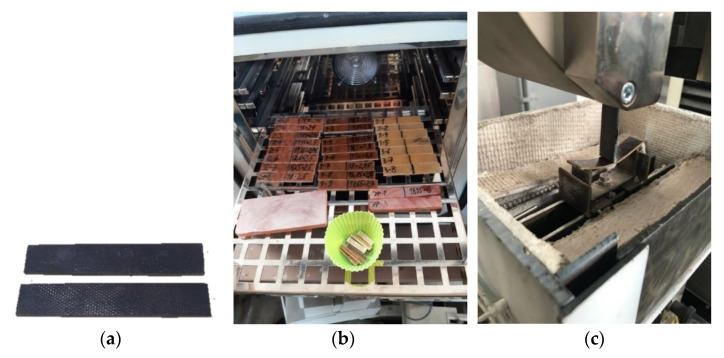
Appearance of fiberglass specimens after prolonged exposure at elevated temperatures (**a**), specimens prepared for exposure (**b**), and three-point bending process at room temperature (**c**).

**Figure 2 polymers-14-01712-f002:**
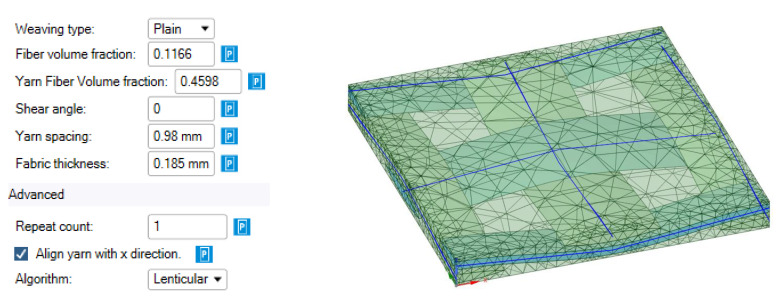
Initial data for constructing geometry, geometric and finite element RVE models.

**Figure 3 polymers-14-01712-f003:**
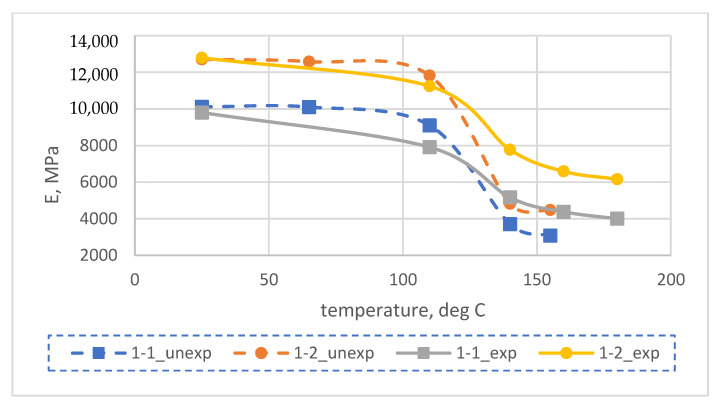
Experimental temperature dependence of elastic modulus (E) at bending of samples No. 1-1, 1-2 of epoxy fiberglass plastic based on EZ-200 glass fabric before (unexp) and after (exp) their exposure to temperatures exceeding the initial glass transition temperature.

**Figure 4 polymers-14-01712-f004:**
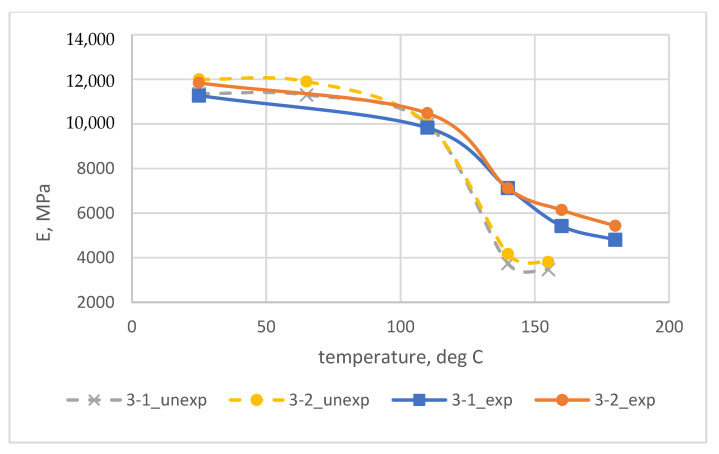
Experimental temperature dependence of elastic modulus (E) at bending of samples No. 3-1, 3-2 of epoxy fiberglass plastic based on T-23 glass fabric before (unexp) and after (exp) their exposure to temperatures exceeding the initial glass transition temperature.

**Figure 5 polymers-14-01712-f005:**
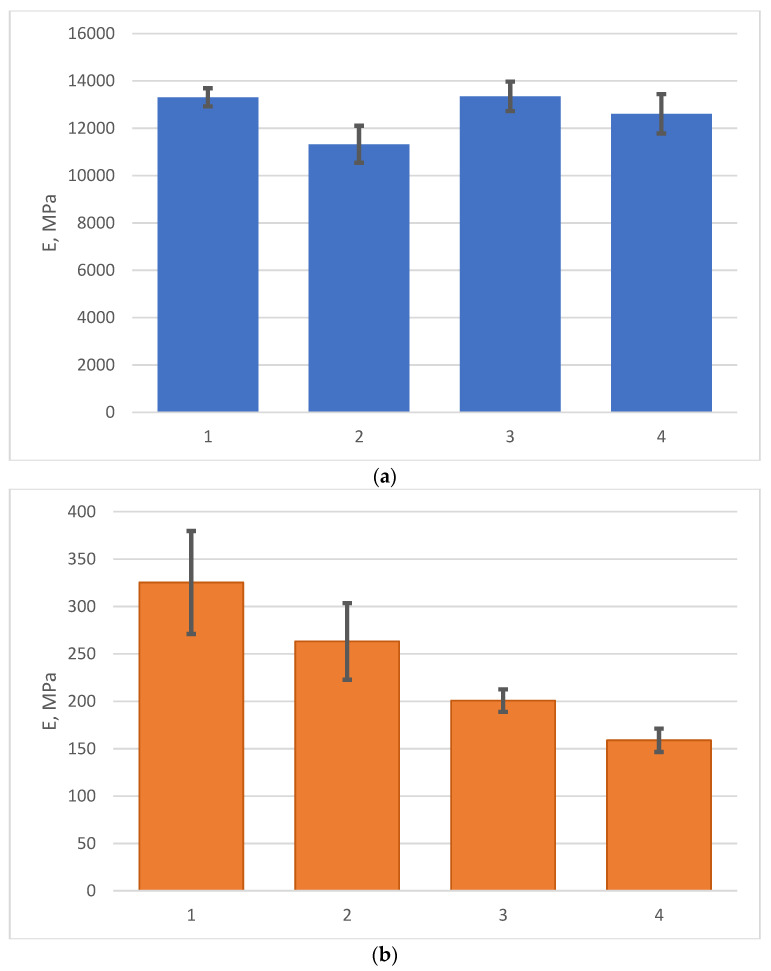
Mean values of elastic modulus (**a**) and bending strength (**b**) with standard deviations: (1) Epoxy FRP on EZ-200 unexposed; (2) Epoxy FRP on EZ-200 exposed; (3) Epoxy FRP on T-23 unexposed; (4) Epoxy FRP on T-23 exposed.

**Figure 6 polymers-14-01712-f006:**
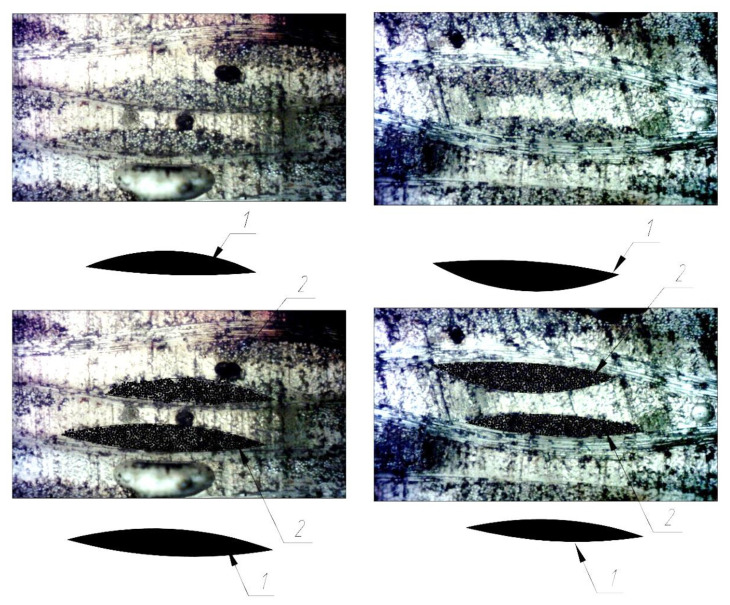
Microphotograph of a fragment of the sample cross-section: (1) highlighted “gross” area of the longitudinal yarn; (2) approximate distribution of fibers inside the yarn.

**Figure 7 polymers-14-01712-f007:**
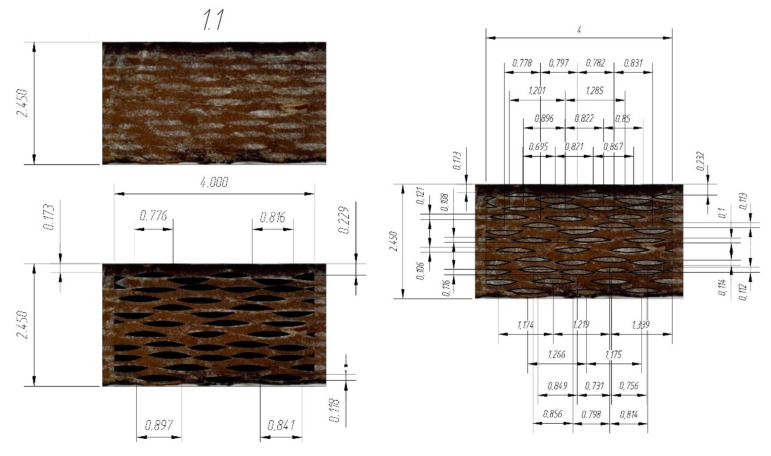
Microphotograph of a cross-section of a fiberglass specimen with actual distances between centers of longitudinal yarns.

**Figure 8 polymers-14-01712-f008:**
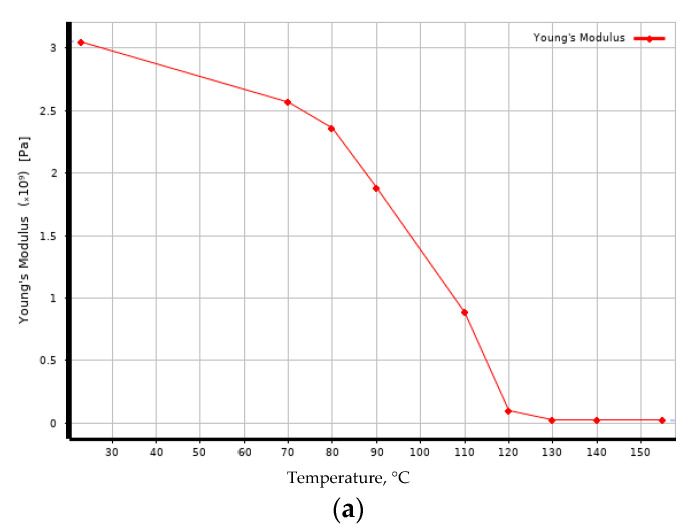
Dependence of elastic modulus of epoxy binder on temperature: (**a**) before exposure; (**b**) after exposure.

**Figure 9 polymers-14-01712-f009:**
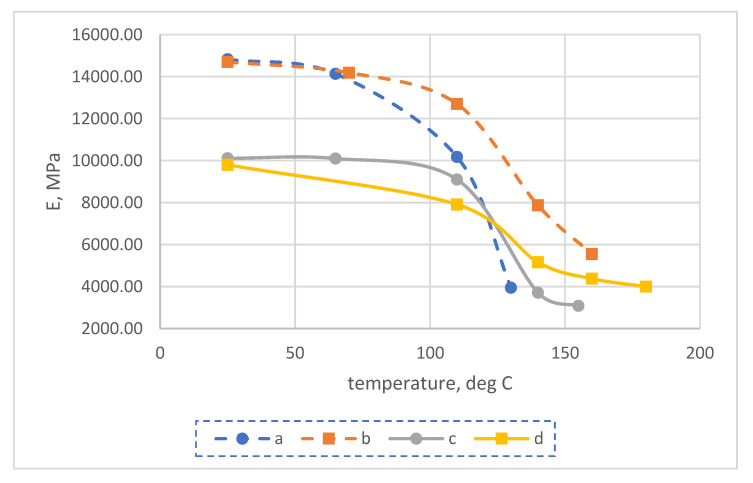
Dependence of elastic modulus on temperature: (a) calculated for the monolayer before exposure at temperature; (b) calculated for the monolayer after soaking at temperature; (c) experimental for the fiberglass sample before exposure at temperature; (d) experimental for the fiberglass sample after aging at the temperature.

**Figure 10 polymers-14-01712-f010:**
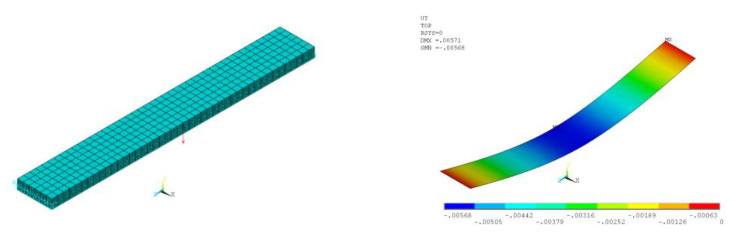
Finite element model of three-point bending of a fiberglass specimen.

**Figure 11 polymers-14-01712-f011:**
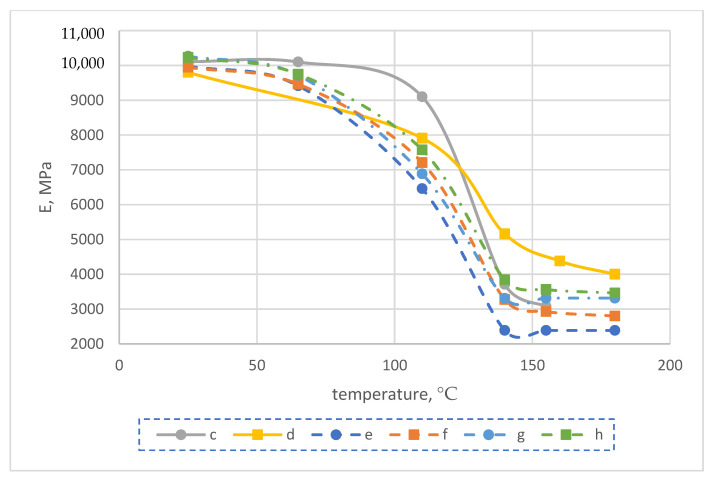
Dependence of elastic modulus on temperature: (c) experimental for the fiberglass specimen before exposure at high temperature; (d) experimental for the fiberglass specimen after exposure at high temperature; (e) computational for the linear calculation of a multilayer composite before exposure at high temperature; (f) computational for the linear calculation of a multilayer composite after exposure at high temperature; (g) computational for the nonlinear calculation of a multilayer composite before exposure at high temperature; (h) computational for the nonlinear calculation of a multilayer composite after exposure at high temperature.

**Figure 12 polymers-14-01712-f012:**
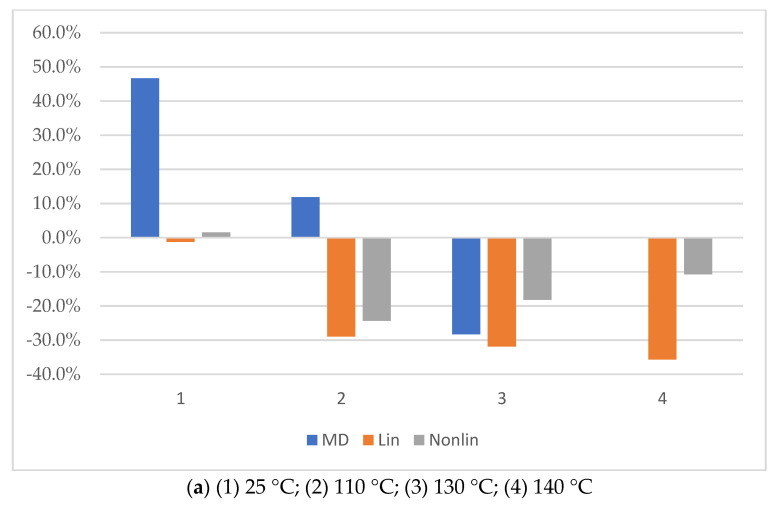
Percent error to the experimental value of the modulus of elasticity of FRP ((**a**) before thermal aging, (**b**) after thermal aging) for the predicted value obtained from the results: (MD) simulation in Material Designer for a reinforced monolayer; (lin) for the linear solution of a multilayer composite bending in ANSYS APDL; (Nonlin) for the nonlinear solution of a multilayer composite bending in ANSYS APDL.

**Figure 13 polymers-14-01712-f013:**
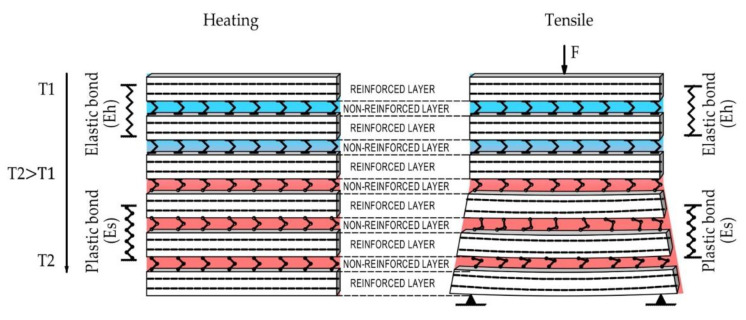
Layer model of deformation of FRP under load when heated.

**Figure 14 polymers-14-01712-f014:**
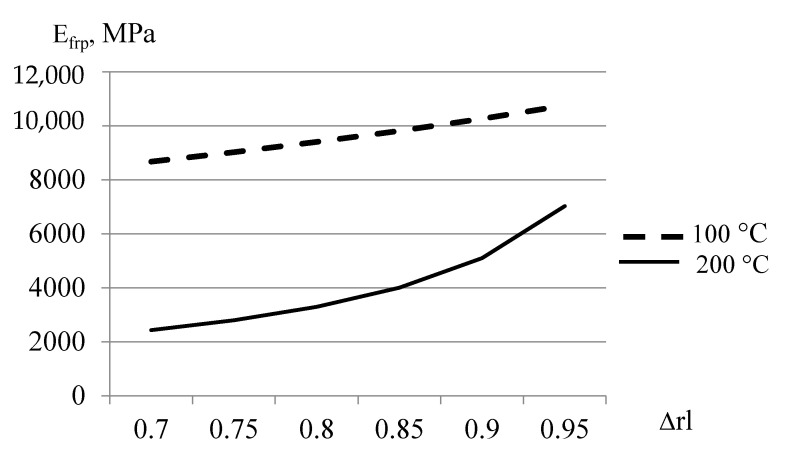
Dependence of elastic modulus of heat-treated fiberglass plastic on the proportion of reinforced layers in the thickness of the plastic at 100 °C and 200 °C.

**Table 1 polymers-14-01712-t001:** Results of bending strength and modulus of elasticity at room temperature of fiberglass plastics before and after curing.

1	2	3	4	5	6	7	8	9	10	11
Description of FRP	Type No	Specimen No	b, mm	h, mm	E, MPa	σ, MPa	E_av_, MPa	σ_av_, MPa	E_unexp_/E_exp_ *	σ_unexp_/σ_exp_ *
Epoxy FRP on EZ-200 unexposed	1	10	17.35	2.3	13.827	286	13,307/381 (2.86%)	325/54 (16.7%)	1.18	1.24
	1	11	18.8	2.4	13.281	374				
	1	12	16.55	2.35	13.380	270				
	1	13	16.85	2.55	12.650	374				
	1	14	18.1	2.35	13.270	272				
	1	15	17.15	2.5	13.435	376				
Epoxy FRP on EZ-200 exposed	1	4	17.75	2.45	12.100	304	11,323/781 (6.90%)	263/40.4 (15.35%)		
	1	5	18.5	2.9	10.150	254				
	1	6	17.6	2.6	10.700	278				
	1	7	18	2.45	12.100	216				
	1	8	16.85	2.55	11.610	308				
	1	9	17.65	2.7	11.280	219				
Epoxy FRP on T-23 unexposed	3	9	17.6	2.45	13.540	205	13,348/622 (4.66%)	201/11.9 (5.92%)	1.06	1.26
	3	10	17.7	2.25	13.160	186				
	3	11	17.55	2.2	13.590	196				
	3	12	16.9	2.45	12.240	193				
	3	13	17.7	2.2	14.100	220				
	3	14	18.65	2.35	13.460	205				
Epoxy FRP on T-23 exposed	3	2	16.35	2.5	11.600	146.4	12,612/831 (6.59%)	159/12.4 (7.77%)		
	3	4	18.6	2.45	11.790	147				
	3	5	18.6	2.45	12.480	150				
	3	6	17.5	2.3	13.720	170				
	3	7	17.75	2.35	13.310	172.3				
	3	8	18.45	2.45	12.770	168				

* E_unexp_/E_exp_ and σ_unexp_/σ_exp_ are the ratios of the modulus of elasticity and bending strength of FRPs before (unexp) and after (exp) exposure at elevated temperatures, respectively.

**Table 2 polymers-14-01712-t002:** Density changes and weight loss of tested fiberglass specimens after exposure.

Description of FRP	Type No	Average Density Unexposed, kg/m^3^	Average Density Exposed, kg/m^3^	Average Density Change	Average Mass Change
Epoxy FRP on EZ-200	1	1607.7	1672.2	4.01%	−0.983%
Epoxy FRP on T-23	3	1803.7	1841.0	2.07%	−1.175%

**Table 3 polymers-14-01712-t003:** Comparative actual and calculated data of the modulus of elasticity of fiberglass plastics on epoxy resin and fiberglass fabric EZ-200 at heating before and after thermal relaxation.

Composition	Temperature, °C	E_fact_, MPa	S, J/J	$k_{pl}$	E_calc_, MPa	% Deviation
EP + EZ-200 before thermal relaxation	25	11.405	2.4	1.000	-	-
	65	11.340		0.698	10.131	−10.7
	110	10.460		0.398	7925	−24.2
	140	4265		0.217	5568	30.6
	160	3780		0.103	3241	−14.3
EP + EZ-200 after thermal relaxation	25	11.295	2.0	1.000	-	
	110	9575		0.498	8741	−8.7
	140	6465		0.347	7311	13.1
	160	5485		0.253	6081	10.9
	180	5080		0.162	4526	10.9
	200	n/a		0.076	2496	-

**Table 4 polymers-14-01712-t004:** Comparative actual and calculated data of the modulus of elasticity of fiberglass plastics on epoxy resin and glass fabric T-23 when heated before and after thermal relaxation.

Composition	Temperature, °C	E_fact_, MPa	S, J/J	$k_{pl}$	E_calc_, MPa	% Deviation
EP + T-23 before thermal relaxation	25	11.680	2.42	1	-	-
	65	11.600		0.695	10.362	−10.7
	110	10.000		0.393	8063	−19.4
	140	3955		0.210	5589	41.3
	160	3640		0.096	3124	−14.2
EP + T-23 after thermal relaxation	25	11.555	2.0	1	-	-
	110	10.155		0.498	8942	−11.9
	140	7115		0.347	7479	5.1
	160	5780		0.253	6221	7.6
	180	5115		0.162	4630	−9.5
	200	n/a		0.076	2553	-

## Data Availability

The data presented in this study are available on request from the corresponding authors.

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
