# Peer review of "Effect of Long-Term Thermal Relaxation of Epoxy Binder on Thermoelasticity of Fiberglass Plastics: Multiscale Modeling and Experiments"

_polymers, 2022, doi:10.3390/polym14091712_

Round 1

Reviewer 1 Report

The manuscript "Effect of Long-Term Thermal Relaxation of Epoxy Binder on Thermoelasticity of Fiberglass Plastics: Prognosis and Experiments" shows extensive studies in modulus and effect above glass transition temperature with included modeling. The work in general is interesting but the authors look mostly from mechanical aspects while the scientific part goes a bit missing. 

  1. In the introduction there need to be more balanced references how other research made and such citation in line 39 with [1-11] should be avoided. It would be much better showing how others solved this issue with given examples. Please include such.
  2.  The Figure 1 what information gives such chimney? Maybe transfer it to supplementary its a bit confusing in this research paper.. In fact it would be much more beneficial showing the chemical structure of the epoxy resin.
  3. The characterization of the material ist missing so please add some FTIR studies as it would be interesting how those condition applied effecting the chemical structure.
  4. The main results are modulus determination in dependence of temperature of the fiberglass specimens. In fact would it not be interesting looking as well on chemical properties how does such changes above glass temperature inflecting in structure? How does as example the hydrophobicity change (as example doing contact angle measurements). 
  5. Figure 6. The micrograph doesn't show so much change or at least that figures difficult to read out. SEM images if there any change appears might be fit much better. Please include those
  6. Figure 4 and 5, Figure 9-12 shows the same method. The manuscript also difficult to read maybe a more compact structure using same method in some section would help to understand the protocol much better. As well the results shown in Tables do not contain standard deviation, did you do only one sample or several. Please include main values with standard deviation to verify reproducibility of your results.
  7.  Table 3 and Table 4 has some Russian language. Please translate those to englisch
  8. The discussion is well written and give some clarity of the results aimed for. There are as well missing parts to other works. Please include those

Author Response

Dear reviewer, thank you for your careful consideration of our work. My co-authors and I have tried to take your comments into account.

Reviewer 2 Report

The paper is of interest. However several improvements are necessary.

The abstract does not reflect the exact methodology adopted.

The RVE method, software, boundary conditions should be specified more precisely.

The error of prediction should be stated in each figure.

Figure 6 has no scale. Is it necessary?

What are numbers/dimensions in fig. 7.

Why are the temperatures show negative (-) sign in fig. 14.

The conclusion is too brief. It should summarize the major inferences from the research.

Overall the data presented does not show statistical significance (error bars). How many times were the sample tested for DMA or TGA? Please specify the details in methodology section. 

Enhance the presentation of figures- with better text size and font.

Define all parameters presented in figures.

Author Response

(The authors gave the same response as above.)

Round 2

Reviewer 1 Report

The authors answered all questions. The manuscript can be considered for publication

Reviewer 2 Report

Can be accepted